# CARING: Cannula for Alleviation of Retinal Injury Caused by Needle Fluidic Gashing

**DOI:** 10.3390/bioengineering11070718

**Published:** 2024-07-15

**Authors:** Kaersti L. Rickels, Anthony L. Gunderman, Mattie S. McLellan, Muhammad M. Shamim, Joseph A. Sanford, Sami H. Uwaydat

**Affiliations:** 1Jones Eye Institute, University of Arkansas for Medical Sciences, Little Rock, AR 72205, USA; klmclellan@uams.edu (K.L.R.); shamim.ophthalmology@gmail.com (M.M.S.); 2Department of Mechanical Engineering, University of Arkansas, Fayetteville, AR 72701, USA; algunder@uark.edu (A.L.G.); mattiemcl3@gmail.com (M.S.M.); 3Institute for Digital Health & Innovation, University of Arkansas for Medical Sciences, Little Rock, AR 72205, USA; jasanford@uams.edu

**Keywords:** cannula tip design, iatrogenic retinal breaks, fluidic gashing, infusion breaks

## Abstract

Infusion-related iatrogenic retinal breaks (IRBs) are a significant complication in vitrectomies, particularly when smaller-gauge cannulas are used during fluid infusion. Using two-dimensional finite element analysis (FEA), we analyzed forces exerted on the retina from different cannulas: traditional 25-gauge, 20-gauge, 23-gauge, and 27-gauge, then investigated four alternative new cannula designs: (A) oblique orifices, (B) external obstruction, (C) side ports, and (D) perpendicular orifices. The analysis revealed that the standard 25-gauge cannula had a force of 0.546 milli-Newtons (mN). Optimized cannulas demonstrated decreased forces: 0.072 mN (A), 0.266 mN (B), 0.417 mN (C), and 0.117 mN (D). While all the designs decrease fluid jet force, each has unique challenges: Design A may complicate manufacturing, B requires unique attachment techniques, C could misdirect fluid toward the lens and peripheral retina, and D requires a sealed trocar/cannula design to prevent unwanted fluid ejection. These four innovative cannula designs, identified with detailed engineering simulations, provide promising strategies to reduce the risk of IRBs during vitrectomy, bridging the gap between engineering insights and clinical application.

## 1. Introduction

Pars plana vitrectomy (PPV) is the third most common ophthalmic surgery performed in the United States, with approximately 225,000 operations performed annually [1,2,3]. Indications for PPV include retinal detachment, macular discomfort [branes, and diabetic vitreous hemorrhage [4]. In the past two decades, PPV instrumentation improvements have primarily focused on reducing instrumentation size for current state-of-the-art systems (from 20-gauge to 27-gauge) [3]. This reduction has enabled sutureless self-sealing wounds, decreased ocular trauma and inflammation, reduced operating times, hastened postoperative recovery, and reduced patient discomfort [5,6,7]. Despite improved instrumentation, intraoperative damage to ocular structures still occurs [8,9,10]. This includes macular hole enlargement [11] and iatrogenic retinal breaks (IRBs).

Iatrogenic retinal breaks are one of the most serious complications following PPV [10]. The formation of an IRB is commonly caused by excessive traction on vitreous strands by intraocular instruments or, even more rarely, when intraocular instruments accidentally contact the retina directly. However, another cause of mechanical injury comes from fluid infusion during air–fluid exchange during PPV [12,13]. During infusion, the high-speed stream leaving the infusion cannula directly strikes the opposite retinal wall, resulting in a strong, localized force that can damage the retina. This is supported by case reports describing retinal injury at the quadrant of the eye where the infusion fluid stream strikes [13,14]. Interestingly, despite the benefits associated with smaller-gauge cannulas, injury rates have been noted to be higher with the 25-gauge PPV instrumentation as compared to the 20-gauge PPV instrumentation [15]. Bilgin et al. found in a series of 492 eyes that underwent 25-gauge surgery that IRBs were observed in 29 of the cases (5.9%) [15]. Three of these twenty-nine cases (10.3%) were caused by the infusion fluid stream. Rishi et al. [12] reported four cases of infusion-related retinal breaks in 25-gauge PPV. Table 1 summarizes cases of infusion fluid stream-related IRB reported in the literature [11,12,13,15].

Reported cannula modifications that aim to reduce the risk of injury caused by the infusion fluid stream during PPV include bent infusion cannulas, cannulas with a blocked infusion lumen at the distal tip, and cannulas with four unblocked lumens cut perpendicular to the longitudinal axis [16,17]. A bent infusion cannula would redirect the fluid stream, but the change in the magnitude of the applied force would likely be insignificant. In the cannula with lumens perpendicular to the longitudinal axis, potential damage may occur to other surrounding structures in the eye such as the lens and the peripheral retina.

In this work, we propose (i) four different infusion cannula tip designs that could reduce the force applied by the infusion fluid stream on the retina during PPV, (ii) an analytical model for evaluating the force of nominal cannula designs, (iii) implementation and validation of finite element analysis (FEA) using Ansys (a simulation modeling environment) for estimating the applied force of our new cannula designs, and (iv) FEA evaluation of the four different infusion cannulas and their comparison to a nominal 25-gauge cannula. The four proposed cannula designs are suitable for use in a standard PPV trocar and include (A) a cannula with a blocked distal tip and ports at an oblique angle to the longitudinal axis of the cannula (oblique orifices), (B) a cannula with a small obstruction in front of the distal tip (external obstruction), (C) a cannula with a blocked distal tip and ports perpendicular to the longitudinal axis of the cannula (side-ports), and (D) a cannula with orifices within, and outside of, the trocar body to disperse the flow (perpendicular orifices).

## 2. Materials and Methods

### 2.1. Fluidic Analytical Modeling

In the nominal infusion cannulas, we can assume laminar flow with a low Reynold’s number of <2000 (as shown in Table 2); thus, the force applied by a uniform infusion fluid stream can be written as the mass flow rate of the fluid multiplied by the fluid’s velocity [18,19,20,21]. This force is written as follows:(1)$$F=\dot{m}u\;$$
where F is the magnitude of the force applied by the infusion fluid stream in Newtons [N], $\dot{m}$ is the mass flow rate of the fluid in kilograms per second $\left[\frac{\mathrm{kg}}{s}\right]$, and u is the magnitude of the fluid speed in meters per second $\left[\frac{m}{s}\right]$. Using the relationship between the mass flow rate and fluid speed with a known cross-section, the mass flow rate can be written as a function of fluid speed using the following equation: (2)$$\dot{m}=\rho uA$$
and the force applied can be written as follows: (3)$$F=u^{2}\rho A$$
where *ρ* is the density of the infusion fluid stream in kilograms per meters cubed $\left[\frac{\mathrm{kg}}{m^{3}}\right]$ and A is the cross-sectional area of the infusion fluid stream at the point of contact in meters squared $\left[m^{2}\right]$. In the clinical application, a volumetric flow rate in milliliters per minute $\left[\frac{\mathrm{mL}}{\min}\right]$ is often used. Thus, Equation (2) can be rewritten in terms of volumetric flow rate using the following relation: (4)$$u=\frac{Q}{A\cdot 60,000}\;$$
where Q is the volumetric flow rate in milliliters per minute $\left[\frac{\mathrm{mL}}{\min}\right]$. We consider the infusion fluid stream of the cannula to be equal to the cross-sectional diameter of the cannula’s inner diameter due to the laminar flow characteristics and the short travel distance of the infusion fluid stream. Therefore, Equation (3) can be rewritten using Equation (4) as follows: (5)$$F=\frac{Q^{2}\rho }{A_{c}\cdot 3.6\times 10^{9}}$$
where $A_{c}$ is the cross-sectional area of the inner diameter of the cannula in meters squared $\left[m^{2}\right]$. Equation (5) presents an analytical solution for evaluating nominal cannula designs based on volumetric flow rate, which will be used to analyze forces exerted on the retina during fluid infusion from nominal cannulas, as well as to validate our proposed FEA model.

### 2.2. Fluidic FEA Simulation 

While Equation (5) provides an analytical solution for the applied force of nominal cannula designs, it is not suitable for analyzing our proposed cannula designs as they significantly change the flow direction, effective fluid density, and fluid cross-sectional area. Here, we propose a method for analyzing the infusion fluid stream force applied by the proposed cannula designs using an FEA model. The FEA model analyzes the fluid dynamics of the liquid-to-air infusion process using a two-dimensional (2-D), transient Eulerian multiphase (water/air) model in the Ansys Fluent environment (Ansys Inc., Canonsburg, PA, USA). The 2-D simulation provides a suitable method for comparing multiple different cannula designs by analyzing the volume fraction of water and the fluid speed of each FEA element. To analyze the force information in 3-D form from the 2-D FEA model, we consider the following: The cross-sectional area of the infusing fluid stream is assumed to have a circular cross-section with a diameter that is equivalent to the width, w, of the infusing fluid stream in 2-D, as shown in Figure 1A,B.The total effective applied force, F, is the cumulative sum of the distributed force, W, induced by the infusing fluid stream across the area defined by the width, w. Note that F and W are assumed to have the same direction as u, written as $\hat{u}=\frac{u}{u}$, where u is the magnitude of u.The density of the infusing fluid stream striking the retinal wall is equal to the density of water $\left(\rho_{\;}=1000\left[\frac{\mathrm{kg}}{m^{3}}\right]\right)$ multiplied by the water volume fraction provided by the FEA model.

Using the above assumptions, and noting that $W=u^{2}\;\hat{u}\cdot \rho \cdot f$, the force applied to the retinal wall can be written as follows:(6)$$F=\int_{A}WdA=u^{2}\;\hat{u}\cdot \rho \cdot f\cdot \pi {\left(\frac{w}{2}\right)}^{2}$$
where f is the volume fraction of water and u, f, and w are the FEA outputs. Note that Equation (6) is written in vector form. Since the directions of all the vectors are parallel, as depicted in Figure 1A,B, we can write (6) in its scalar form, similar to (1)–(5), as follows:(7)$$F=\int_{A}WdA=u^{2}\cdot \rho \cdot f\cdot \pi {\left(\frac{w}{2}\right)}^{2}$$

The FEA setup uses 2-D geometry that consists of a 24 mm circle representing the eye and a set of concentric tubes, one representing the trocar and one representing the cannula. The cannula extends 6 mm into the eye and the distal end of the trocar is located 4 mm proximal to the cannula tip. The cannula and trocar geometry are modified based on the cannula size (i.e., 20-gauge, 23-gauge, 25-gauge, and 27-gauge cannulas). Additionally, the 25-gauge cannula geometry is further modified based on our intuition to simulate our proposed improved design. The simulation setup consists of a mesh of square cells, each with a dimensional width of 0.1 mm (Figure 2A). The boundary conditions of the simulation include a constant inlet boundary condition of water with a fluid speed corresponding to Equation (4), and a no-slip wall boundary condition for the water. The internal pressure of the eye is set to 35 mmHg. Initialization is performed using Ansys’s hybrid initialization procedure, and the surface is patched to ensure the initial phase is air (Figure 2B). Computations are performed using a time step of 5 × 10^−4^ s for 0.101 s, ensuring the fluid has sufficient time to contact the wall and distribute along the retinal surface. Following the computation, the volume fraction, fluid speed, and width of the impinging jet are obtained a single time step after the infusion fluid stream contacts the retinal surface. This time step is selected due to the assumption that as the fluid fills the eye, the infused fluid dampens the impact; thus, the localized force is most relevant after initial contact.

## 3. Results

### 3.1. Analytical Evaluation of Nominal Cannulas 

Using Equation (5), a quantitative comparison is made between the cannula ID (gauge size) and the force of the infusion cannula fluid stream. A constant volumetric flow rate of 10 $\left[\frac{\mathrm{mL}}{\min}\right]$ is used for comparison (Figure 3). Note that if the volumetric flow rate remains the same (i.e., the pressure drop across a smaller (higher gauge) cannula is compensated to maintain the volumetric flow rate), the force applied by the infusion fluid stream (see Equation (3)) increases with a smaller cannula. This is due to the increasing fluid speed needed to maintain the volumetric flow rate through a smaller cross-sectional area (see Equation (4)). Increasing gauge size results in a smaller PPV port size and smaller wounds but may increase the likelihood of complications associated with the infusion fluid stream. Reducing the size of the infusion cannula from 20-gauge to 23-, 25-, and 27-gauge increases the magnitude of the force of the infusion fluid stream by a factor of 3.173, 5.336, and 8.184, respectively. Reducing the size of the cannula from 23-gauge to 25- or 27-gauge increases the unit force of the fluid jet by a factor of 1.682 and 2.579.

### 3.2. FEA Evaluation of Nominal Cannulas

Using the analytical force values generated from Equation (5) for the 20-, 23-, 25-, and 27-gauge cannulas in the prior section, a comparative study was used to validate our FEA model and the applicability of Equation (7). This comparison can be seen in Table 3. The error between the force values using Equation (5) and the force values generated from the FEA model and Equation (6) was 3.56 ± 0.9%. This suggests that our model is accurate and can be used for cannula development and comparison. An example of the 20-gauge cannula FEA model and the 25-gauge cannula FEA model can be seen in Figure 4A,B and Figure 4C,D, respectively. Figure 4A,C depict the water volume fraction contours while Figure 4B,D depict the velocity contours. Video simulation of the infusion fluid stream for standard 25 g cannula can be seen in our Appendix A.

### 3.3. Alternative Cannula Designs 

The proposed cannula designs were compared to a standard 25-gauge cannula using the proposed and validated FEA model. Methods of improving cannula performance by reducing the force applied by the infusion fluid stream include (A) adding orifices at an oblique angle to the cannula’s longitudinal axis with the distal tip completely blocked (oblique orifices), (B) making the distal end appear more similar to a spark plug by adding a small external obstruction at the distal tip of the cannula (external obstruction), (C) adding side ports to the cannula tip perpendicular to the longitudinal axis with the distal tip completely blocked (side-ports), and (D) adding orifices perpendicular to the longitudinal axis of the cannula at the cannula tip and along the cannula body (perpendicular orifices). Video simulations of the infusion fluid stream for alternative cannula designs can be seen in our Appendix A. Appendix A cannula with oblique orifices, Appendix A external “spark plug” obstruction, Appendix A two side ports with the tip of the cannula occluded, and Appendix A multiple orifices along the length of the cannula.

The velocity contours of the cannula designs with the infusion fluid stream contour superimposed on it can be seen in Figure 5. Note that each design can reduce the force applied to the retinal wall, as shown in Table 4. Compared to the standard 25-gauge cannula, the respective percent decreases in force on the retina with each cannula design are as follows: (A) 87%, (B) 51%, (C) 24%, (D) 79%.

## 4. Discussion

Iatrogenic retinal breaks are a serious complication during PPV. Our literature review showed that this complication is more frequently encountered with smaller cannulas. Using the derivations above, we show the force applied by the infusion fluid stream can be equated to the density of the fluid multiplied by the volumetric flow rate squared and divided by the area of the stream. We note that the area of the stream can be assumed to be the inner cross-sectional area of the cannula in the nominal cannula configuration. Thus, the force applied by a jet stream is proportional to the inverse of the area, or inverse of the radius squared [20,21]. Consequently, by reducing the radius by a multiple of 2.32 (ratio of 20-gauge cannula inner diameter to 25-gauge cannula inner diameter, i.e., 0.603/0.26 [mm/mm]), the force is increased by a multiple of 5.336 if the flowrate in the two cannulas remains the same. This likely explains the higher incidence of IRBs in 25-gauge vitrectomies compared to 23- or 20-gauge vitrectomies [22,23,24]. Additionally, when considering the shape of the jet stream from smaller gauge cannulas, this directed force on the opposing retina is the likely culprit of the observed IRBs. When observing the jet streams’ shapes and force of the alternative cannulas in comparison, the flow force of (A) was slowed down significantly, (B) and (D) spread the flow stream out into two, less forceful streams, and (C) directed the streams perpendicular to the longitudinal axis of the cannula. These alternatives move the jet force away from the macula, sparing central vision damage. A jet with reduced flow force is less likely to cause an IRB. 

While current efforts toward improving instrumentation have focused on reducing instrument size, in this study, we proposed four alternative cannula designs and developed a 2-D fluidic FEA model to examine the force applied to the retina during fluid–air exchange in vitreoretinal surgery. The goal was to determine if these alternative cannula tip designs could reduce the amount of force applied to the retina during surgery, thus reducing the risk of iatrogenic retinal breaks. To reduce the effective force exerted by the fluid, we considered designs that distribute the force imparted by the fluid across a larger area. These designs accomplish this objective by introducing additional possible flow paths. Considering the percent decrease of force applied to the retina, and the minor limitations to design A, it is likely that this cannula, with oblique angle orifices, would be the best at increasing flow paths, distributing the energy applied to the retina while minimizing theoretical damage to alternative structures in the eye as compared to both standard cannulas and the alternative cannulas.

While each alternative cannula design reduces the force applied to the retinal wall, each also has limitations. Design A generates a fluid stream with the lowest force and would be the safest. There are potential manufacturing difficulties, such as complicated fixture designs and material removal operations. For multiport design (A), (Figure 5A), the oblique angle of 60° of the orifices would pose difficulties in manufacturing. The external obstruction design (B), which appears like a spark plug (Figure 5B), requires the inclusion of an obstruction that would require unique attachment techniques. The side port design (C) (Figure 5C) alternatively directs the fluid towards the lens of the eye and peripheral retina, risking damage. The final design (D), with perpendicular orifices (Figure 5D), requires a sealed trocar/cannula design to prevent fluid from ejecting from the proximal end of the trocar. 

Despite these potential limitations, each design presents an engineering novelty that can reduce the risk of retinal damage caused by smaller inner diameter cannulas while enabling sutureless operations that are possible using 25-gauge cannulas. 

## Figures and Tables

**Figure 1 bioengineering-11-00718-f001:**
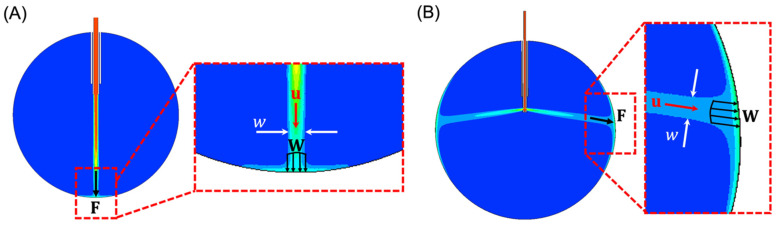
(**A**) FEA result of the water volume fraction using a 20-gauge cannula system with a 24 mm eye. In the detailed view (inset), the width parameter w (white arrows) can be seen depicted. The water jet has a velocity u (red), that exerts a distributed force, w (four black arrows). This distributed force produces a total effective force F (single black arrow). Blue represents 0% water (i.e., air) and red represents 100% water. (**B**) FEA result of the water volume fraction with orifices perpendicular to the longitudinal axis of the cannula.

**Figure 2 bioengineering-11-00718-f002:**
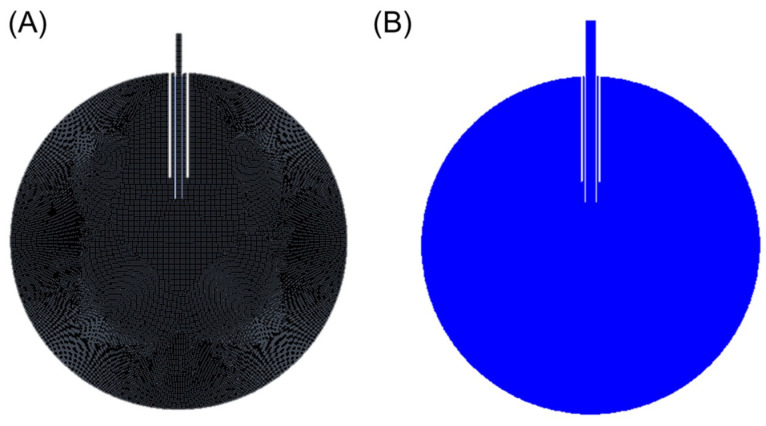
(**A**) Mesh resulting from a cell size of 1e-4 m. (**B**) Resulting volume fraction contour after initialization and patch (blue represents air in this figure).

**Figure 3 bioengineering-11-00718-f003:**
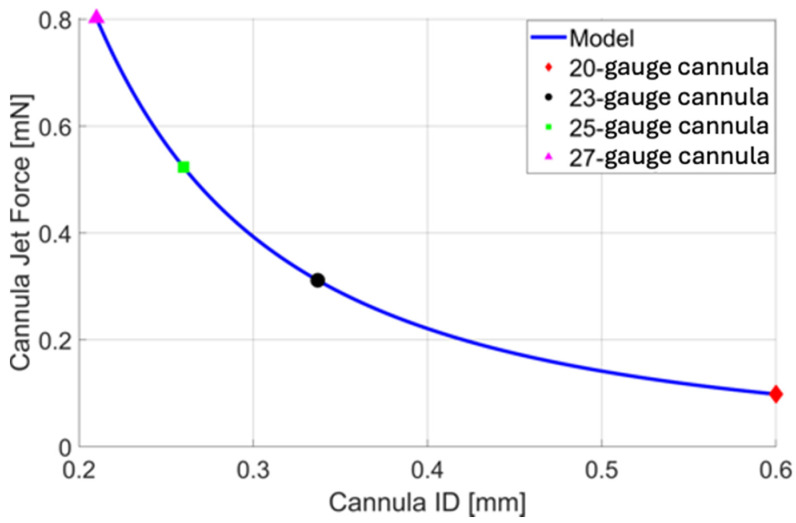
A plot of cannula infusion fluid stream force vs. cannula inner diameter (ID) based on Equation 5 with a volumetric flow rate of 10 $\left[\frac{\mathrm{mL}}{\min}\right]$. For quick reference, the force resulting from the infusion fluid stream of a 20-gauge (red), 23-gauge (black), 25-gauge (green), and 27-gauge (magenta) cannula is presented, allowing us to assess their nominal cannula designs.

**Figure 4 bioengineering-11-00718-f004:**
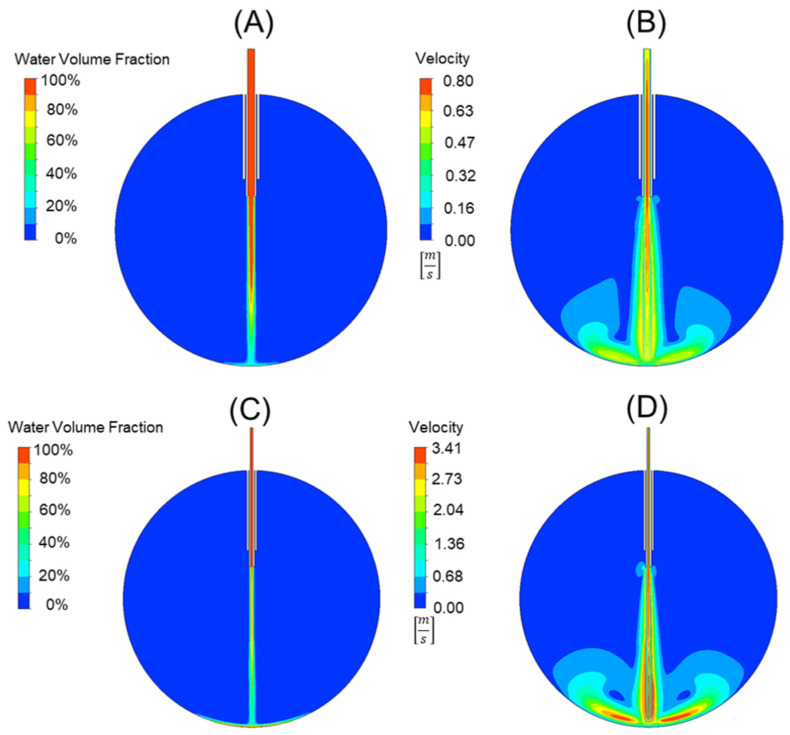
A comparison of a 20-gauge cannula (**A**,**B**) to a 25-gauge cannula (**C**,**D**). The water volume fraction contours (**A**,**C**) can be seen alongside the velocity contours (**B**,**D**). Note that the scales are not the same for the velocity plot, with red representing 0.78 [m/s] in (**B**) and 3.41 [m/s] in (**D**).

**Figure 5 bioengineering-11-00718-f005:**
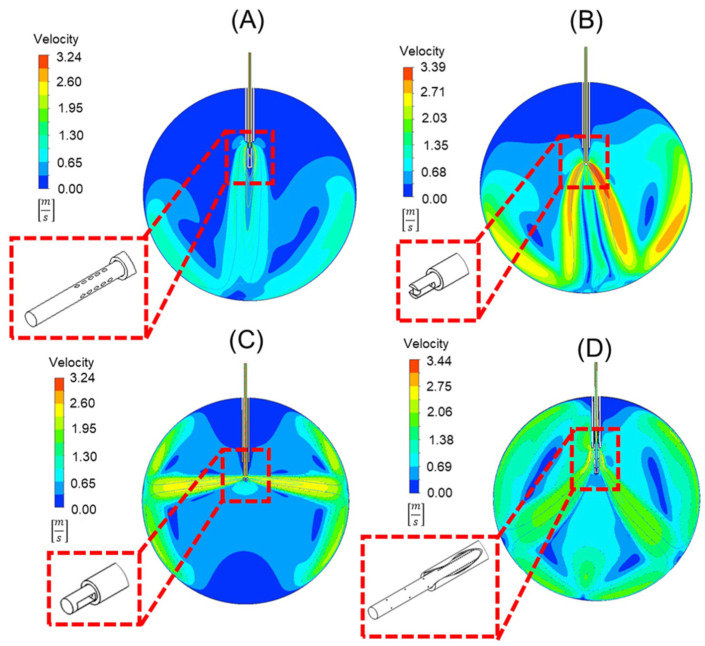
Velocity contours of four different cannula designs. These include (**A**) oblique orifices at an angle of 60°, (**B**) an external obstruction similar to a spark plug, (**C**) orifices perpendicular to the longitudinal axis of the cannula, and (**D**) a design that uses orifices in the cannula within, and outside of, the trocar body to disperse flow. Depicted in each is a detailed view of the geometry of the trocar and cannula in 3-D.

**Table 1 bioengineering-11-00718-t001:** Cases of fluid infusion-related retinal damage following pars plana vitrectomy reported in the literature.

Paper	Patient	Procedure	Complication
Rishi et al. [12]	Case 1	CEIOL * + 25 G PPV ^†^ for FTMH ^‡^	Retinal break
	Case 2	25 G PPV for FTMH	Retinal break > RRD ^§^
	Case 3	PPV for FTMH	Retinal break
	Case 4	25 G PPV for FTMH	Retinal break > RRD
Bilgin et al. [15]	Case 1	25 G PPV for vitreous hemorrhage	Retinal break
	Case 2	25 G PPV for silicone oil extraction, ERM ^||^ removal	Retinal break > RRD
	Case 3	PPV for silicone oil extraction	Retinal break > RRD
Belenje et al. [13]	Case 1	25 G PPV for FTMH	Retinal break > RRD
Zacharias et al. [11]	Case 1	23 G PPV for macular hole repair	Enlargement of macular hole

Abbreviations: CEIOL *: cataract extraction with intraocular lens implantation. PPV ^†^: pars plana vitrectomy. FTMH ^‡^: full-thickness macular hole. RRD ^§^: rhegmatogenous retinal detachment. ERM ^||^: epiretinal membrane.

**Table 2 bioengineering-11-00718-t002:** Reynold’s number for nominal cannulas with 10 [mL/min] volumetric flow rate.

Needle	ID [mm]	OD [mm]	Reynold’s Number
25-gauge	0.26	0.52	917
20-gauge	0.60	0.91	395
23-gauge	0.34	0.64	707
27-gauge	0.21	0.41	1135

**Table 3 bioengineering-11-00718-t003:** Analytical vs. model force calculations of nominal cannulas.

Gauge	20	23	25	27
Analytical Force [mN]	0.098	0.311	0.523	0.802
Model Force [mN]	0.096	0.323	0.546	0.834
Error [%]	2.04%	3.85%	4.39%	3.99%

Abbreviations: mN = milli-Newtons.

**Table 4 bioengineering-11-00718-t004:** Force values of optimized cannulas.

	Figure 4D	Figure 5A	Figure 5B	Figure 5C	Figure 5D
Force [mN]	0.546	0.072	0.266	0.417	0.117

Abbreviations: mN = milli-Newtons.

## Data Availability

The original contributions presented in the study are included in the article/Appendix A. Further inquiries can be directed to the corresponding author/s.

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
