# Peer review of "CARING: Cannula for Alleviation of Retinal Injury Caused by Needle Fluidic Gashing"

_bioengineering, 2024, doi:10.3390/bioengineering11070718_

Round 1

Reviewer 1 Report

Comments and Suggestions for Authors

The authors designed four cannulas to decrease the fluid jet force exerted on the retina. The mechanics of water infusion from the cannula to the eye are simulated using commercial finite element analysis (FEA) software. The four designs are evaluated using the FEA method. The flow analysis and FEA simulation seem fine. But there are some questions regarding the cannula design:

1.     The authors claim that the cannula design is novel. But it seems that it is easy to think of a “new” energy-dissipating design that reduces the kinetic energy of water flow in a pipe. What are the authors’ considerations and novelty in designing these structures? This part should be described in detail.

2.     Have these designs been optimized? Which one among the four is the best, in your opinion? Will the design with the lowest force perform best in clinical situations? Any further in vitro or ex vivo experiments to validate your claims and findings.

3.     Although the starting point of this study is a complication in clinical practice, the result and discussion part of this article are not quite relevant to clinical treatment. I would appreciate it if the authors could add more discussion regarding the effect of different flow situations in the eye, to improve the relevance between the new cannula design and its potential clinical impact (like reducing retinal damage).

Reviewer 2 Report

Comments and Suggestions for Authors

please find attached

Round 2

Reviewer 1 Report

Comments and Suggestions for Authors

The authors have addressed the questions and now can be accepted.